# Reliability of the retrospective Clinical Interview Schedule Revised (rCIS-R) to assess relapse in depression in primary care patients

**Larisa Duffy**[1]*, **Louise Marston**[2,3], **Gemma Lewis**[1], **Glyn Lewis**[1]

**1** Division of Psychiatry, University College London, London, United Kingdom, **2** Research Dept. of Primary Care and Population Health, University College London, London, United Kingdom, **3** Priment Clinical Trials Unit, University College London, London, United Kingdom

* larisa.duffy@ucl.ac.uk

## Abstract

### Objectives

We are not aware of a simple and short structured measure that retrospectively assesses time to relapse for depression. We developed the retrospective Clinical Interview Schedule Revised (rCIS-R) to assess depression relapse in the previous 12 weeks, for use in a clinical trial of maintenance antidepressant treatment. We assessed test-retest reliability and construct validity in relation to a Global Rating Question (GRQ) about worsening mood, participants stopping their study medication and Patient Health Questionnaire (PHQ-9) scores.

### Methods

In our study 444 participants provided data for rCIS-R, GRQ and PHQ-9 and 396 participants completed rCIS-R on two occasions about 30 minutes apart. The reliability study was nested within a randomised controlled trial (ANTLER).

### Results

We found substantial test-retest agreement for the rCIS-R definition of relapse (kappa 0.84 (95%CI 0.71 to 0.97)), for individual sections and timing of relapse (Intraclass Correlation Coefficient 0.94 (95%CI 0.92 to 0.95)). Comparison of relapse with GRQ, stopping study medication and PHQ-9 supported the construct validity of the rCIS-R.

### Conclusions

The rCIS-R provides a reliable way of assessing relapse of depression over the previous 12 weeks. Its brevity, self-report format, simplicity of scoring and absence of training requirement makes it attractive to use in randomised controlled trials.

**Data Availability Statement:** The study participants consented to their data being "looked at only by authorized members of the Sponsor, research team, regulatory authorities and the NHS

trust," and this was approved by the ethics board. The authors are therefore unable to share the data publicly. However, the data will be provided to researchers via a collaboration with the ANTLER research team. To gain access, researchers can sign a data access agreement with the study sponsor (priment@ucl.ac.uk, University College London, London, UK) and data can then be made available with support from the ANTLER research team.

**Funding:** The ANTLER trial was independent research commissioned by the National Institute for Health Research (NIHR) Health Technology Assessment (HTA) Programme 13/115/48 The funding source had no role in study design, data collection, data analysis, interpretation or writing of this paper. The corresponding author had full access to all data used in the study, and final responsibility for the decision to submit for publication. We acknowledge the support of the UCLH BRC.

**Competing interests:** GL received fees from fortitude Law in relation to litigation on antidepressant withdrawal symptoms. This does not alter our adherence to PLOS ONE policies on sharing data and materials. There are no patents, products in development or marketed products associated with this research to declare.

## Introduction

Depression is a leading cause of disability, with more than 300 million people having depressive illness worldwide [1]. A substantial proportion of the burden of depression arises from relapses, recurrence and chronicity [2]. Depression is usually treated in primary care [3] and the treatment choices are informed by the results of clinical trials. Therefore, an accurate assessment of the reappearance of depressive symptoms is needed if researchers are to study interventions designed to reduce future episodes.

One of the methodological challenges of measuring relapse in depression studies has been an aspiration to distinguish relapse from recurrence using the "gold standard" definitions set up by Frank [4] and further developed by Rush [5]. However, their approach has proved challenging due to lack of clarity between the terms, dependency on the measure used and requirements for frequent (i.e. fortnightly) assessments. Such an approach is often impractical and the definitions themselves are loosely justified empirically. Evidence from longitudinal studies suggests that depression should no longer be seen as a time-limited disorder with episodes lasting around four to six months but rather thought of as a "relapsing-remitting" continuum with debilitating symptoms occurring between acute episodes [6]. We believe that studies assessing the benefit of interventions need to measure the reappearance of any depressive symptoms and that the distinction between relapse and recurrence is less important. We therefore use the term relapse in this manuscript to refer to any new reappearance of depressive symptoms.

Another issue with assessing relapse of depression is the limitations of scales currently used in clinical trials because they either assess a short period of time or require clinical experience to administer. To measure relapse, most studies have used rating scales administered by clinicians, such as the Hamilton Rating Scale for Depression (HRSD) [7] or Montgomery and Äsberg Depression Rating Scale (MADRS [8]) or self-administered assessments such as the Beck Depression Inventory (BDI) [9] and Patient Health Questionnaire (PHQ-9) [10]. To measure relapse, studies have repeated such measures at frequent intervals, typically fortnightly. This introduces additional participant burden and increases the expense of studies. Even though these measures assess current symptoms, the time to relapse can be estimated if the assessments are given frequently enough.

Fully structured interviews have also been used in population based longitudinal studies investigating the course of depression. They eliminate observer bias and can be administered by lay interviewers, so are more economical. An example is the Composite International Diagnostic Interview (CIDI) [11], which could in principle be used in clinical trials. However, the interview has over 280 symptom questions that are accompanied by probe questions to assess severity. The CIDI is extremely lengthy and difficult to administer and can take up to 3 hours, increasing the burden on participants and reducing its acceptability. In addition, rigid rules of administration and the use of complex flow charts may lead to mistakes by the interviewer in either presenting questions or interpreting participants' responses [12]. As far as we are aware it has never been used in clinical trials and would be unsuitable in its full form though restricting it to the section on depression would make it more feasible. The Structured Clinical Interview Disorder (SCID) [13] is a semi-structured interview intended to be administered by trained diagnosticians, which can be expensive. It is lengthy in its full form (between 2 and 6 hours) and requires judgements about the presence of symptoms so observer bias could be introduced. The inter-rater reliability of SCID has produced fair agreement on depression [14] using audiotapes that would have overestimated the reliability as it does not include variation as a result of the interviewer. SCID, in part, relies on interviewers generating their own

questions so even though it has satisfactory reliability when interviewers are trained together, the reliability across different studies or over time is not known.

A more efficient, pragmatic way of assessing relapse of depressive symptoms after recovery in clinical trials might be a simple and short fully-structured self-administered questionnaire, asking about depressive symptoms in a retrospective way over the past several weeks which does not require extensive staff training. We are not aware of any assessments that fit this description.

To that end, we adapted the Clinical Interview Schedule—Revised version (CIS-R) [15], a validated measure that has been widely used to assess severity and duration of depression. The CIS-R asks about symptoms in the last 7 days; our modified version which we called the retrospective CIS-R (rCIS-R) assesses symptoms in the last 12 weeks, in a fully structured format. The aim of this study was to assess the test-retest reliability of rCIS-R. We also investigated the construct validity of rCIS-R in relation to the Global Rating Question (GRQ) (i.e. patients reporting feeling worse), patients stopping their study medication and PHQ-9 as a depression severity measure.

## Methods

### Study design and participants

Our study was a part of the ANTidepressants to prevent reLapse in dEpRession (ANTLER) study [16]. Our study was a reliability study within this multicentre, pragmatic, double blind individually randomised parallel group-controlled trial that was registered with ISRCTN (ISRCTN15969819). The trial was approved by the National Research Ethics Service committee, East of England—Cambridge South (ref: 16/EE/0032). Clinical trial authorisation was granted by the Medicines and Healthcare Products Regulatory Agency (MHRA). All participants provided written informed consent. The trial protocol [17] is published in full, but in brief: participants were recruited from 150 primary care practices at four UK sites: London, Bristol, Southampton and York. Patients were identified via database searches or during consultation and were eligible if they were aged 18 to 74, had experienced at least two episodes of depression; had been taking antidepressants for nine months or more but were well enough to consider stopping medication. Exclusion criteria were current depression according to ICD-10 at baseline, comorbid psychiatric disorder, inability to complete the questionnaires in English, major alcohol or substance abuse. The trial compared continuing with one of citalopram 20mg, sertraline 100mg, fluoxetine 20mg or mirtazapine 30mg with replacement of the medication with an identical placebo after a tapering period of one month for fluoxetine or two months for the other medication. The randomisation was minimised by the four study sites, the four medications and severity of depressive symptoms at baseline (two categories measured using the CIS-R). The trial intervention was for 52 weeks and participants were followed up at 6, 12, 26, 39 and 52 weeks. The baseline and all follow-ups bar 6 weeks were face-to-face with a researcher. The 6-week follow-up consisted of a postal questionnaire. The primary outcome of the main trial was time to relapse of depression, assessed by rCIS-R at 12, 26, 39 and 52 weeks. The results of the main trial have already been published [16].

Our aim was to assess the reliability and to explore construct validity of the measure used to assess the main outcome. For the purpose of the reliability study, participants were asked to complete the rCIS-R twice approximately 30 minutes apart, at one of the follow-up appointments. The follow-up appointment began with the first completion of rCIS-R then the participant spent 30 minutes on paper and pen questionnaires and completion of computerised tasks and then finished with the second completion of the rCIS-R. For the purpose of this paper, we

treat participants who completed the rCIS-R twice as a single cohort regardless of group allocation or follow-up timing.

## Measures

The rCIS-R was a modified version of CIS-R and designed as a self-administered computerised questionnaire and asked about the previous 12 weeks [18]. The 12 weeks recall period was chosen because the follow-up appointments were spaced at roughly 3 months or 12 week intervals and it was convenient anchor point for participants to remember what has happened since they last met with the researcher. Also, the 12 week interval was considered as appropriate length for participants to remember. Of note, the follow ups in the trial were about 13 weeks apart on average. Five sections (depressive mood, depressive ideas, concentration, sleep and fatigue) were used to assess symptoms and were asked along with questions about duration of the symptoms, their intensity during the worst week and when the symptom/s began. The rCIS-R starts with two overarching mandatory questions for the depressive mood and depressive ideas sections. If the participant's answers to the two questions indicated that they had experienced either low mood or anhedonia in the last 12 weeks, they were asked about duration, to establish that symptoms had been present for at least two weeks and the time they started feeling depressed. If the symptom/s were present for two weeks or longer, the participant was considered positive for that symptom and was asked 10 additional questions covering depressive symptoms during the worst week in the previous 12 (e.g. feeling low for prolonged periods, unresponsiveness of mood, loss of sexual interest, restlessness, suicidal thoughts, etc the full list of themes of questions are in the S1 File).

The other three sections (concentration, sleep and fatigue) of the rCIS-R were similar in structure; they also start with mandatory question/s. If the participant's answer to the mandatory question/s indicated that they had not experienced such symptoms, then the extra questions relating to severity of the symptom were not asked and the participant skipped to the mandatory question of the next section. If the participant's answer indicated that they had have the symptom, they were considered positive for that section and asked further questions about their experience during the worst week. It was possible to score a maximum of three on the concentration and fatigue sections and four on the sleep section.

Fig 1 provides the concentration section as an example of a section from rCIS-R: the first two questions are mandatory and if the answer is "yes" to at least one, the other three questions are asked.

The assessment takes approximately five minutes to complete, though if the participant does not have any symptoms then it takes as little as two minutes.

Relapse was defined as experiencing two or more depressive symptoms from any of the five sections during the worst week in the past three months (this must include at least one of the two overarching mandatory questions on depressive mood or anhedonia for at least two weeks). We also defined relapse in line with ICD-10 criterion and investigated the number of participants experiencing four or more depressive symptoms. In addition to defining a binary outcome of relapse, rCIS-R generates a total score for the depressive episode that occurred in the previous 12 weeks. Each section generates a maximum score between three and six; higher scores indicate more symptoms and the total score can range from zero to 21.

At baseline, 12, 26, 39 and 52 weeks, participants also completed the Global Rating Question (GRQ): "Compared to when we last saw you, how have your moods and feelings changed?" Responses were: 'I feel a lot better'; 'I feel slightly better'; 'I feel about the same'; 'I feel slightly worse'; 'I feel a lot worse'. We created a dichotomous variable: feeling worse (1) and feeling the same or better (0).

"Has there been a period of time in the PAST THREE MONTHS, when you had problems in concentrating on what you were doing?"

"Has there been a period of time in the PAST THREE MONTHS, when you noticed any problems with forgetting things?"

"During the worst WEEK in the past three months:

1. Could you concentrate on all of the following without your mind wandering?
   A whole TV programme
   A newspaper article
   Talking to someone"

   "Yes, I could concentrate on all of them"
   "No, I couldn't concentrate on at least one of these things"

2. Did problems with your concentration STOP you from getting on with things you used to do or would like to do?"

   "No"
   "Yes"

3. Did you forget anything important?"

   "No"
   "Yes, I did forget something important"

**Fig 1. Concentration section from rCIS-R.**

At each time point, participants completed the PHQ-9, a nine-item self-administered questionnaire. Each of the nine DSM-IV items have four response options ranging from "0" (not at all) to "3" (nearly every day). Total scores range from zero to 27. If there were one or two scores missing, we replaced the score by the mean of the scores present. If there were more than two items missing, we considered the questionnaire missing for that participant.

## Statistical methods

Level of agreement between first and second completion of rCIS-R was assessed using kappa (quadratic weighted and unweighted) statistics. Quadratic weighted and unweighted kappa produced very similar results. Weighted kappa provides a ratio-scale degree of disagreement

to each cell of the k x k table and is more suitable as a measure of agreement. We also used weighted kappa to assess agreement on time of depression relapse and performed sensitivity analyses investigating the agreement (i) within a younger (18 to 56 years old) and an older (57 to 74 years old) sample; and (ii) agreement by education group. We considered these variables could be related to the ability of the participant to remember their previous responses.

The level of agreement between two completions was also assessed using the methods described by Bland–Altman [19]. To assess both, agreement and degree of correlation, we calculated the Intraclass Correlation Coefficient (ICC) using a single-measurement, absolute-agreement, 2-way mixed-effects model. To assess construct validity, we conducted exploratory analysis of whether people who relapsed (i) stopped their study medication either due to returning to medication prescribed by their GP outside the trial or to withdrawing from the trial, (ii) reported they were worse on the GRQ at 12 weeks. In addition, we analysed the association between rCIS-R scores (as the outcome) and PHQ-9 scores at 12 weeks using linear regression modelling. All analyses were conducted using STATA 14.

## Results

### Sample characteristics

Out of 478 participants recruited in the trial, from them 396 completed rCIS-R twice at a given time point, i.e. at any of the four follow-ups. Our intention was for all 478 participants to complete the rCIS-R twice at 26 week follow-up appointment for consistency and simplicity of the fieldwork. However, if participants preferred to complete it at different follow-up appointment, we allowed it. As a result, two participants completed rCIS-R twice at 12-week follow-up appointment, 335 at 26 weeks, 42 at 39 weeks and 17 at 52 weeks. There were 106 males; the mean age was 55 (SD 6) years and 6% of participants reported being from an ethnic minority. Thirty-seven percent of participants had a degree, 32% were educated to A level and 31% had GCSE or other education. London site recruited 43% of the sample, Bristol 20%, Southampton 21% and York 16%. At enrolment into the study, 46% of the sample were on Citalopram, 34% on Fluoxetine, 16% on Sertraline and 4% on Mirtazapine. The mean PHQ-9 score was 3.8 (SD 3.6) (Table 1).

Kappa (k) for relapse in depression was 0.84 (95%CI 0.71 to 0.97) indicating substantial agreement between the first and second completion of rCIS-R. The level of agreement of the individual sections of rCIS-R was also substantial (Table 2).

Twenty percent of participants met relapse criteria at the first completion (n = 80) and at second completion that was done by the same participants 30 minutes later by 19% (n = 77). The mean score of the first completion of rCIS-R was 6.67, SD 5.06; the mean score of second completion was 6.41, SD 5.25. The mean total score difference was -0.25 (95%CI -0.43 to -0.07). The ICC was 0.94 (95%CI 0.92 to 0.95).

The agreement on timing of depression relapse was substantial on month with weighted k = 0.84 (95%CI 0.71 to 0.97) and week k = 0.87 (95%CI 0.74 to 1.00) of reappearance of depression.

The results of a sensitivity analysis investigating the agreement (i) within a younger (18 to 56 years old) and an older (57 to 74 years old) sample; and (ii) agreement by education group are presented in Table 3.

Eighteen percent of participants met the threshold for relapse according to the ICD-10 diagnostic criteria for depression at the first (n = 72) and second (n = 70) completions. Kappa for relapse in depression according to ICD-10 was 0.74 (95% CI 0.64 to 0.84).

The Bland-Altman plot in Fig 2 shows the agreement between first and second completion of rCIS-R. The central red line (-0.25) is the mean difference of the 1st and 2nd completion

**Table 1. Baseline characteristics of the sample who completed rCIS-R twice at the same follow-up appointment.**

| Characteristic | (Mean) or n with characteristic (N = 396) | (SD) or % |
|---|---|---|
| Age, (mean) | (55) | (6) |
| Male | 106 | 27 |
| Ethnicity | | |
| White | 373 | 94 |
| Ethnic minority | 23 | 6 |
| Highest educational qualification | | |
| Degree/ higher degree | 146 | 37 |
| Diploma/ A Levels or equivalent | 127 | 32 |
| GCSE* or equivalent/ other/ none | 123 | 31 |
| Site | | |
| London | 170 | 43 |
| Bristol | 80 | 20 |
| Southampton | 84 | 21 |
| York | 62 | 16 |
| Antidepressant | | |
| Sertraline | 62 | 16 |
| Citalopram | 183 | 46 |
| Fluoxetine | 135 | 34 |
| Mirtazapine | 16 | 4 |
| PhQ-9 score at baseline, (mean) | (3.8) | (3.6) |
| Age first became aware of having depression, (mean) | (32) | (5) |
| Time point when completed rCIS-R twice | | |
| 12 week follow-up | 2 | 0.5 |
| 26 week follow-up | 335 | 85 |
| 39 week follow-up | 42 | 11 |
| 52 week follow-up | 17 | 4 |

* The General Certificate of Secondary Education (GCSE) is an academic qualification, generally taken in a number of subjects by pupils in secondary education in England, Wales and Northern Ireland at age 16 (end of compulsory schooling)

scores, the other two red lines (-3.9 and 3.4) are two standard deviations above and below the mean difference. Most of the observations lay within the two SD (with 4.7% (n = 19) out) and are randomly scattered over the lengths of the scale, which indicates no bias between 1st and 2nd completion of the rCIS-R, nor with increasing score.

**Table 2. Level of agreement for relapse in depression, individual symptoms, between 1st and 2nd completion of retrospective CIS-R.**

| | Frequency (%) present at 1st completion | Weighted kappa | 95%CI |
|---|---|---|---|
| Relapse | 80 (20) | 0.84 | 0.71 to 0.97 |
| Symptoms | | | |
| Depressive mood | 214 (54) | 0.87 | 0.77 to 0.97 |
| Depressive thoughts | 200 (50) | 0.87 | 0.77 to 0.87 |
| Fatigue | 222 (56) | 0.85 | 0.75 to 0.95 |
| Concentration | 100 (25) | 0.81 | 0.72 to 0.91 |
| Sleep | 206 (52) | 0.91 | 0.82 to 1.00 |

**Table 3. Level of agreement for relapse in depression between 1ˢᵗ and 2ⁿᵈ completion of retrospective CIS-R: Numbers and kappa.**

| | neg1*, neg2 | neg1, pos2 | pos1, neg2 | pos1, pos2 | Kappa (95% CI) |
|---|---|---|---|---|---|
| Relapse in all participants | 301 | 15 | 18 | 62 | 0.84 (0.71 to 0.97) |
| Relapse (ICD10) | 310 | 14 | 16 | 56 | 0.74 (0.64 to 0.84) |
| Stratified by Age (n) | | | | | |
| 18 to 56 (205) | 159 | 5 | 9 | 32 | 0.78 (0.64 to 0.92) |
| 57 to 74 (191) | 142 | 10 | 9 | 30 | 0.70 (0.56 to 0.84) |
| Stratified by Education (n) | | | | | |
| degree and higher (146) | 106 | 6 | 5 | 29 | 0.79 (0.63 to 0.95) |
| A level or equivalent (127) | 102 | 3 | 6 | 16 | 0.74 (0.57 to 0.92) |
| GCSE and lower (121) | 91 | 6 | 7 | 17 | 0.66 (0.48 to 0.83) |

*neg1 = did not relapse at 1ˢᵗ completion; neg2 = did not relapse at 2ⁿᵈ completion; pos1 = relapsed at 1ˢᵗ completion; pos2 = relapsed at 2ⁿᵈ completion

The results of exploratory analysis in Table 4 show that the almost two thirds of participants who had relapsed by 12 weeks' follow-up reported feeling worse (63%). The odds of experiencing feeling worse were over five times greater for those who relapsed than for those who did not as measured by rCIS-R at 12 weeks (OR: 5.55; 95% CI: 3.44 to 8.95; $p < 0.0001$).

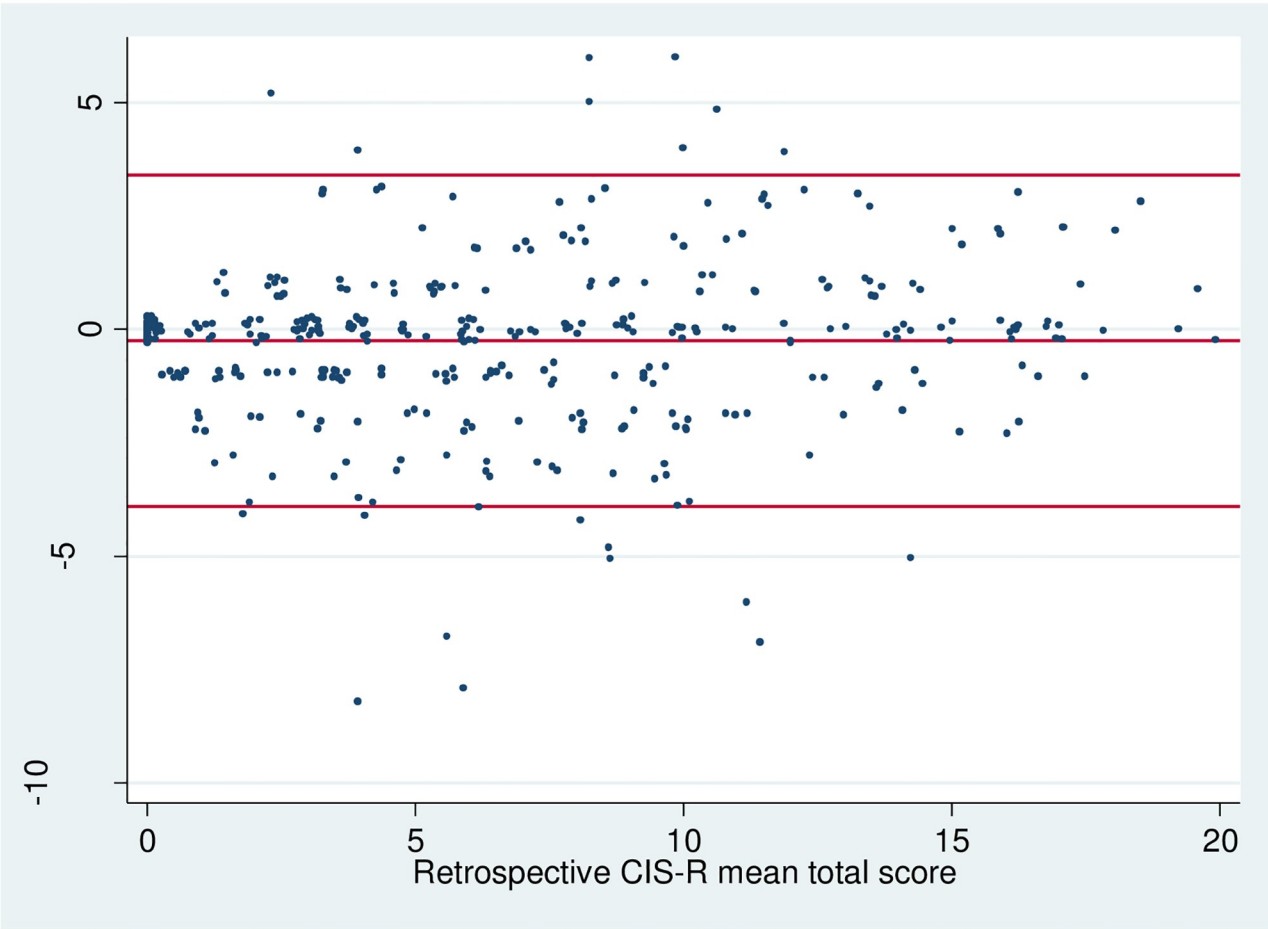

**Fig 2. Bland-Altman plot showing the difference in total rCIS-R score against mean.**

**Table 4. Association of relapse measured by rCIS-R and self-reporting feeling worse (GRQ) at 12 weeks.**

| Relapse by 12 weeks | Global rating question at 12 weeks | | Total |
|---|---|---|---|
| | Feeling worse | Feel the same or better | |
| Did not relapse n (%) | 80 (23) | 265 (77) | 345 (100) |
| Relapsed n (%) | 62 (63) | 37 (37) | 99 (100) |
| Total n (%) | 142 (32) | 302 (68) | 444 (100) |

**Table 5. Number of participants (%) who went back onto their usual antidepressants or stayed on trial medication or were not on any medication at 12 weeks when they relapsed.**

| Relapse by 12 weeks | Medication at 12 weeks | | | Total |
|---|---|---|---|---|
| | Still on trial medication | Back on usual medication | Not taking trial or usual medication | |
| Did not relapse n (%) | 337(95) | 10 (3) | 6 (2) | 353(100) |
| Relapsed n (%) | 67 (69) | 19 (20) | 11 (11) | 97 (100) |
| Total n (%) | 404 (90) | 29 (6) | 17 (4) | 450 (100) |
| P-value | | | | <0.0001 |

**Table 6. Linear regression analysis of association between PhQ9 score at 12 weeks and rCIS-R score at 12 weeks.**

| | n | Coefficient | 95%CI | Standardised regression coefficient | P value |
|---|---|---|---|---|---|
| Unadjusted | 444 | 0.83 | 0.76 to 0.90 | 0.73 | <0.0001 |
| Adjusted[a] | 444 | 0.75 | 0.67 to 0.82 | 0.66 | <0.0001 |

[a]Adjusted for minimisation variables (centre, medication, CIS-R score at baseline: above or below median), gender and group allocation variables

Table 5 shows that 20% of those who relapsed, stopped study medication and returned to their usual antidepressant medication by 12 weeks' follow-up, compared to 3% of those who did not relapse.

We found strong evidence of a correlation (0.73) between the PHQ-9 and rCIS-R at 12 weeks. Table 6 shows the linear regression analysis of association between PHQ-9 score at 12 weeks and rCIS-R score at 12 weeks for the 444 participants who completed both. For every unit increase on PHQ-9 score, rCIS-R score increases by 0.83 (95% CI 0.76 to 0.90) in unadjusted and 0.75 (95% CI 0.67 to 0.82) in adjusted linear regression models.

## Discussion

The results of our study nested within the ANTLER trial provide strong evidence that the rCIS-R is a reliable measure for assessing reappearance of depressive symptoms among primary care patients. There was substantial agreement for definitions of relapse, for the individual symptoms that were assessed, for the sum of the symptoms scores and timing of relapse. The association between rCIS-R and GRQ indicated that the relapses identified were likely to be clinically important. The rCIS-R identified two thirds of relapses amongst those who self-reported feeling worse on the GRQ. Construct validity was supported by an association between the total scores of rCIS-R and PHQ-9 scores at 12-week follow-up. The sample size of 396 was large enough to allow the limits of agreement to be estimated precisely. The reliability we report here is similar to the reliability reported for other assessments commonly used in mental health research. However, the SCID reliability study assessed the ratings of two raters

scoring the same interview by means of audio tapes, which may have overestimated results [14]. If instead two separate interviews were compared the results could have provided a more accurate estimate of the reliability of the SCID. The rCIS-R, as a fully-structured measure, does not have such weaknesses as all questions are known, so it is not compromised by variations on interviewing skills and styles of the raters. The kappa values in our study seem more realistic than those of the reliability of CIDI where most kappa values were very high, i.e. 0.9 and over. This could be due to strength of our study design eliminating any possible study design violations.

## Strengths and limitations

The ANTLER sample is likely to be more representative than samples from other prior trials and we think there will be good generalisabiliy of our results to the use of the rCIS-R in other samples. The results of our study are likely to generalise to the adult population on antidepressant medication and considering stopping it, even though ANTLER was a randomised trial sample. Participants in randomised trials tend to be more compliant with treatment compared to general population and in ANTLER only a small proportion of those potentially eligible participated. Furthermore, as not all ANTLER participants took part in the test-retest reliability study this could be further compounded. However, the sensitivity analysis confirmed that although younger and/or better-educated groups had a slightly higher level of agreement, the agreement in older and/or less educated groups was still either substantial or very good.

One problem with testing reliability using the test-retest method is that participants could potentially learn or memorise answers, leading to the first completion influencing the second. A short time interval makes this more likely, whereas a longer interval, e.g. several days or weeks, increases the chances of changes in depressive symptoms leading to an underestimate of the reliability. In our study, multiple questionnaires consisting of numerous items were administered in the 30 minutes between rCIS-R completions, minimising the potential effect of memory. In addition, we found that reliability in younger and/or better educated participants was similar to older and/or less well educated participants, so there was little evidence of any learning or memorising of the answers during the first completion.

The Bland-Altman plot did not provide evidence that participants reported more symptoms at 1st completion of rCIS-R and there was little difference between 1st and 2nd completion.

The results show that rCIS-R is reliable, i.e. it is consistent across repeated measures. Assessing validity of any psychiatric measure is problematic, as a vital component of the validation process is the selection of the appropriate reference method against which to assess the test measurement. There are considerable problems involved with measurement of "true relapse". The rCIS-R uses the same questions as those used in the widely used and validated CIS-R [20] and so this is in itself a good indicator that our new measure is a valid measure of depression, albeit assessed retrospectively. One possible design for our study would have been to repeatedly assess participants every week or so during the 12-week period assessed by the rCISR. We did not have such data and so we estimated construct validity in relation to other relevant outcomes. To support construct validity of the new assessment, we chose three indicators: self-reported global rating of worsening (GRQ), stopping trial medication and PHQ-9, which is the most commonly used assessment of depression in primary care [21]. Assessments aimed at determining current depressive symptoms, such as PHQ-9, may differ from methods aimed at the assessment of past symptoms because the latter are reliant upon the memory and conceptualisation skills of the individual. Therefore, the construct validation method of assessing association of rCIS-R and PHQ-9 on their own may not have been sufficiently robust.

However, our results of a strong association, between rCIS-R and participants feeling worse and stopping medication, support the construct validity of rCIS-R at assessing relapse.

Our sample of primary care patients demonstrated that people appeared very reliable at recalling times when they were depressed over the past 12 weeks. Simplicity has practical advantages: brief reliable measures are more likely to be used in clinical trials and busy primary care settings. There has been some evidence that as little as one or two questions are effective at screening for an acute depressive episode [22, 23]. Similarly, our measure can be used to detect relapses and save resources in clinical trials within busy primary care settings.

## Conclusion

Our study used a novel assessment (rCIS-R) to measure reappearance of depressive symptoms. The results of our test-retest reliability study nested within the ANTLER trial provide strong evidence that the rCIS-R is a reliable and valid measure of assessing the reappearance of depressive symptoms. The main advantage of rCIS-R is that, to our knowledge, it is the only simple and short fully-structured measure assessing time to relapse. Compared to other widely used semi-structured scales, the pragmatic advantages of our measure are the brief time required for completion, simplicity of scoring, and absence of any elaborate special training requirements. Our study also investigated the reliability of the rCIS-R as a diagnostic tool. Though primarily designed for research purposes the rCIS-R may also have application in clinical practice as a simple way of assessing relapse in depression.

## Supporting information

**S1 File. rCIS-R sections.**
(DOCX)

## Acknowledgments

We are grateful to all the patients that took part in the ANTLER trial. We thank the staff in participating general practitioner surgeries for their help with recruitment. We acknowledge the support of the National Institute for Health Research University College London Hospitals Biomedical Research Centre (BRC). The views expressed are those of the authors and not necessarily those of the NIHR, NHS or the Department of Health and Social Care.

## Author Contributions

**Conceptualization:** Glyn Lewis.

**Data curation:** Larisa Duffy.

**Formal analysis:** Larisa Duffy, Louise Marston.

**Funding acquisition:** Glyn Lewis.

**Methodology:** Louise Marston, Gemma Lewis, Glyn Lewis.

**Project administration:** Larisa Duffy.

**Supervision:** Louise Marston, Gemma Lewis, Glyn Lewis.

**Validation:** Larisa Duffy.

**Writing – original draft:** Larisa Duffy.

**Writing – review & editing:** Louise Marston, Gemma Lewis, Glyn Lewis.

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
