## [Decision Letter · Decision Letter 0]

18 Aug 2022

PONE-D-22-01963

Reliability of the retrospective Clinical Interview Schedule Revised (rCIS-R) to assess relapse in depression in primary care patients.

PLOS ONE

Dear Dr. Duffy,

Thank you for submitting your manuscript to PLOS ONE. After careful consideration, we feel that it has merit but does not fully meet PLOS ONE’s publication criteria as it currently stands. Therefore, we invite you to submit a revised version of the manuscript that addresses the points raised during the review process.

Please note that we have only been able to secure a single reviewer to assess your manuscript. We are issuing a decision on your manuscript at this point to prevent further delays in the evaluation of your manuscript. Please be aware that the editor who handles your revised manuscript might find it necessary to invite additional reviewers to assess this work once the revised manuscript is submitted. However, we will aim to proceed on the basis of this single review if possible. 

The reviewer has raised a number of concerns that need attention. They request additional information on methodological aspects of the study (such as outcome measurements), as well as revisions to the data and statistical analyses.

Could you please revise you manuscript to address all their concerns?

We look forward to receiving your revised manuscript.

Kind regards,

Thomas Tischer

Staff Editor

PLOS ONE

Journal Requirements:

2. Please ensure you have included the registration number for the clinical trial referenced in the manuscript.

Furthermore we have noted that your study does not require additional ethics approval since the study was embedded in a clinical trial. However for reporting purposes, please could you include the ethics approval information (as well as information for informed consent) of the original clinical trial. 

4. Thank you for stating the following in the Acknowledgments Section of your manuscript: "We are grateful to all the patients that took part in the ANTLER trial. We thank the staff in participating general practitioner surgeries for their help with recruitment. We acknowledge the support of the National Institute for Health Research University College London Hospitals Biomedical Research Centre (BRC). The ANTLER trial was independent research commissioned by the National Institute for Health Research HTA Programme 13/115/48. The views expressed are those of the authors and not necessarily those of the NIHR, NHS or the Department of Health and Social Care."

Please remove any funding-related text from the manuscript and let us know how you would like to update your Funding Statement. Currently, your Funding Statement reads as follows: "Prof Glyn Lewis received funding to conduct ANTLER trial from the National Institute for Health Research HTA Programme 13/115/48.

NO, The funders had no role in study design, data collection and analysis, decision to publish, or preparation of the manuscript."

5. We noted in your submission details that a portion of your manuscript may have been presented or published elsewhere. "" ext-link-type="uri" xlink:type="simple">https://doi.org/10.3310/hta25690" Please clarify whether this publication was peer-reviewed and formally published. If this work was previously peer-reviewed and published, in the cover letter please provide the reason that this work does not constitute dual publication and should be included in the current manuscript.

7. Please note that in order to use the direct billing option the corresponding author must be affiliated with the chosen institute. Please either amend your manuscript to change the affiliation or corresponding author, or email us at plosone@plos.org with a request to remove this option.

8. We noticed you have some minor occurrence of overlapping text with the following previous publication(s), which needs to be addressed:

- https://eprints.soton.ac.uk/452827/1/3038159.pdf

In your revision ensure you cite all your sources (including your own works), and quote or rephrase any duplicated text outside the methods section. Further consideration is dependent on these concerns being addressed.

Reviewers' comments:

Reviewer's Responses to Questions

**Comments to the Author**

1. Is the manuscript technically sound, and do the data support the conclusions?

Reviewer #1: Partly

2. Has the statistical analysis been performed appropriately and rigorously? 

Reviewer #1: No

3. Have the authors made all data underlying the findings in their manuscript fully available?

Reviewer #1: No

4. Is the manuscript presented in an intelligible fashion and written in standard English?

Reviewer #1: No

5. Review Comments to the Author

Reviewer #1: Line 130: rCIS-r were measured at 12, 26, 39 and 52 weeks.

Line 138: The rCIS-R was …. asked about the previous 12 weeks.

This is confusing. Is the measure all about previous 12 weeks, even though it is measured at 26, 39 and 52 weeks?

Did all 396 participants have outcome measures at every time point?

If rCIS-r, GRQ and PHQ9 were measured at all time points, why did the analysis focus on 12 weeks only?

The primary outcome is defined as time to relapse. Survival analysis by randomization group? No data analysis is performed for this outcome except the reliability by week or month which is not clearly described.

Table 4 and 5, how did you define relapse if there is a disagreement between 1st and 2nd completion?

Table 4 and 5, how about other time points? P value to table 5?

Table 6, correlation coefficient or standardized regression coefficient are better to be reported as HPH9 and rCIS-r have different units. Add p values.

“Excellent” is a too strong word for a kappa of 0.84.

6. PLOS authors have the option to publish the peer review history of their article (what does this mean?). If published, this will include your full peer review and any attached files.

Reviewer #1: No

---

## [Author Response · Author response to Decision Letter 0]

17 Oct 2022

Dear Dr Tischer,

Re: PONE-D-22-01963 

Reliability of the retrospective Clinical Interview Schedule Revised (rCIS-R) to assess relapse in depression in primary care patients.

PLOS ONE

Thank you for the invitation to submit a revised version of our manuscript. We provide a point-by-point response to the editor’s and reviewer’s comments below. Judging by reviewer’s comments, it seems the main criticism stems from confusion between the main trial (ANTLER) and this study. Our manuscript describes a study nested within the ANTLER trial. The purpose of our study was to assess the reliability and validity of the measure that was used in the ANTLER trial to assess relapse in depression. We have made several changes to make clear distinction between our manuscript and other trial papers, so the reader can see that the statistical analysis of our study and its purpose was different from previous publications concerned with the main trial. The changes are punctuated in the response below.

We have also included: 

• A marked-up copy of the manuscript that highlights changes made to the original version. This was upload as a separate file labelled 'Revised Manuscript with Track Changes'.

• An unmarked version of our revised paper without tracked changes, uploaded as a separate file labelled 'Manuscript'.

• The figure files are also resubmitted as .tif files

Yours sincerely,

Larisa Duffy 

UCL Division of Psychiatry

Maple House, 149 Tottenham Court Road

London W1T 7NF

 

Editors’ comments

PLOS ONE

Journal Requirements:

Response: Thank you for providing the PLOS ONE style templates. We have followed them carefully and made several changes to the title, authors’ affiliations and addresses on the title page of the manuscript. 

As per the PLOS ONE’s style requirements, we have added the title of Fig.1 ‘Concentration section from rCIS-R’ (page 8) and Fig.2 ‘Bland-Altman plot showing the difference in total rCIS-R score against mean’ (page 15). 

We also converted the figures from their original format (word) into .tif format and resubmitted them separately.

2. Please ensure you have included the registration number for the clinical trial referenced in the manuscript.

Furthermore we have noted that your study does not require additional ethics approval since the study was embedded in a clinical trial. However for reporting purposes, please could you include the ethics approval information (as well as information for informed consent) of the original clinical trial. 

Response: We have added the following information on the ethics approval of the trial and the information on consent (page 6): “The trial was approved by the National Research Ethics Service committee, East of England - Cambridge South (ref: 16/EE/0032). Clinical trial authorisation was granted by the Medicines and Healthcare Products Regulatory Agency (MHRA). All participants provided written informed consent.” 

We have also included the registration number for the clinical trial, also on page 6: “…trial was registered with ISRCTN (ISRCTN15969819)”

Response: We have removed the ‘Role of the Funding source’ section (page 10). Our new Funding Statement should read: “The ANTLER trial was independent research commissioned by the National Institute for Health Research (NIHR) Health Technology Assessment (HTA) Programme 13/115/48 The funding source had no role in study design, data collection, data analysis, interpretation or writing of this paper. The corresponding author had full access to all data used in the study, and final responsibility for the decision to submit for publication. We acknowledge the support of the UCLH BRC”. 

4. Thank you for stating the following in the Acknowledgments Section of your manuscript: "We are grateful to all the patients that took part in the ANTLER trial. We thank the staff in participating general practitioner surgeries for their help with recruitment. We acknowledge the support of the National Institute for Health Research University College London Hospitals Biomedical Research Centre (BRC). The ANTLER trial was independent research commissioned by the National Institute for Health Research HTA Programme 13/115/48. The views expressed are those of the authors and not necessarily those of the NIHR, NHS or the Department of Health and Social Care."

Please remove any funding-related text from the manuscript and let us know how you would like to update your Funding Statement. Currently, your Funding Statement reads as follows: "Prof Glyn Lewis received funding to conduct ANTLER trial from the National Institute for Health Research HTA Programme 13/115/48.

NO, The funders had no role in study design, data collection and analysis, decision to publish, or preparation of the manuscript."

Response: we have removed the funding information from the Acknowledgments section (page 20). The section now reads: “We are grateful to all the patients that took part in the ANTLER trial. We thank the staff in participating general practitioner surgeries for their help with recruitment. We acknowledge the support of the National Institute for Health Research University College London Hospitals Biomedical Research Centre (BRC). The views expressed are those of the authors and not necessarily those of the NIHR, NHS or the Department of Health and Social Care.” 

Please update the Funding statement to “The ANTLER trial was independent research commissioned by the National Institute for Health Research (NIHR) Health Technology Assessment (HTA) Programme 13/115/48 The funding source had no role in study design, data collection, data analysis, interpretation or writing of this paper. The corresponding author had full access to all data used in the study, and final responsibility for the decision to submit for publication. We acknowledge the support of the UCLH BRC.”

We have also removed funding related text from other areas of the manuscript, i.e. the second sentence in the Methods section on page 6 was changed from: “It was a multicentre, pragmatic, double blind individually randomised parallel group-controlled trial that was funded by commissioned by the National Institute for Health Research (NIHR) Health Technology Assessment (HTA) Programme” to “Our study was a reliability study within this multicentre, pragmatic, double blind individually randomised parallel group-controlled trial that was registered with ISRCTN (ISRCTN15969819).”

5. We noted in your submission details that a portion of your manuscript may have been presented or published elsewhere. "https://doi.org/10.3310/hta25690" Please clarify whether this publication was peer-reviewed and formally published. If this work was previously peer-reviewed and published, in the cover letter please provide the reason that this work does not constitute dual publication and should be included in the current manuscript.

Response: There is some overlap between our manuscript with our report published in November 2021 by the funder in the Health Technology Assessment Volume: 25, Issue: 69. This a common occurrence as the report is a standard request from the funder. The report covers all aspects of the research grant and includes all results including any sub-studies. The report was peer-reviewed; NIHR allow dual publication and we have been publishing papers separately, e.g. trial outcome paper: https://www.nejm.org/doi/full/10.1056/NEJMoa2106356 and health economics paper: https://doi.org/10.1007/s40258-021-00693-x

Response: We would like to provide the following Data Availability statement: “The study participants consented to their data being “looked at only by authorised members of the Sponsor, research team, regulatory authorities and the NHS trust”, and this was approved by our ethics board. We are therefore unable to share the data publicly. However, we will provide the complete dataset to researchers via a collaboration agreement with the ANTLER research team. To gain access, researchers can sign a data access agreement with the study sponsor (priment@ucl.ac.uk University College London, London, UK) and data can then be made available with support from the ANTLER research team.”

7. Please note that in order to use the direct billing option the corresponding author must be affiliated with the chosen institute. Please either amend your manuscript to change the affiliation or corresponding author, or email us at plosone@plos.org with a request to remove this option.

Response: Please note, both the affiliation and billing institution should be “University College London”

8. We noticed you have some minor occurrence of overlapping text with the following previous publication(s), which needs to be addressed:

- https://eprints.soton.ac.uk/452827/1/3038159.pdf

In your revision ensure you cite all your sources (including your own works), and quote or rephrase any duplicated text outside the methods section. Further consideration is dependent on these concerns being addressed.

Response: The link is the same report discussed above (please see Response to point 5). Although the same results are described by us in the trial report and our manuscript, we have rephrased the text as much as possible and we have also cited the report as a reference on page 7 where the measure is described in detail.

Reviewers' comments:

Reviewer's Responses to Questions

Comments to the Author

1. Is the manuscript technically sound, and do the data support the conclusions?

Reviewer #1: Partly 

Response: We have made a number of changes to the manuscript, that are described below and hope they have improved its quality. 

2. Has the statistical analysis been performed appropriately and rigorously? 

Reviewer #1: No 

Response: Judging by this and some other comments from the reviewer, it seems that there has been some confusion between the main trial and this study where the reviewer criticised our analysis as there was an expectation to see the analysis of the primary outcome of the main trial. We are grateful to the reviewer for identifying the lack of clarity in our manuscript. We have made several changes to make the distinction between our paper and other trial papers, so the reader can see that the statistical analysis of our study and its purpose was different from previous publications concerned with the main trial. We are confident that the statistical analyses we have performed in this paper are both appropriate and performed rigorously. 

The main trial, i.e. ANTLER trial has been extensively published:

Trial protocol: https://doi.org/10.1186/s13063-019-3390-8

Trial statistical analysis plan: https://discovery.ucl.ac.uk/id/eprint/10089782/

Trial outcome paper: https://www.nejm.org/doi/full/10.1056/NEJMoa2106356

Health economics paper: https://doi.org/10.1007/s40258-021-00693-x

The current manuscript describes a study nested within the ANTLER trial. The purpose of the study was to assess the reliability and validity of the measure that was used in the ANTLER trial to assess relapse in depression. Although the manuscript mentions that our “study nested within ANTLER trial”, we have made a number of additional changes in the manuscript (pages 6, 7, 10 and 14) to make a clear distinction between the main trial and our study.

3. Have the authors made all data underlying the findings in their manuscript fully available?

Reviewer #1: No

Response: The data will be made available to any researcher wishing to collaborate with the study team. This is a requirement in line with the consent from ANTLER participants. To gain access, researchers will need to sign a data access agreement with the study sponsor (priment@ucl.ac.uk, University College London, London, UK). We are unable to upload the data due to ethical restrictions. 

We suggest the following Data Availability statement: “The study participants consented to their data being “looked at only by authorised members of the Sponsor, research team, regulatory authorities and the NHS trust”, and this was approved by the ethics board. We are therefore unable to share the data publicly. However, we will provide the complete dataset to researchers via a collaboration with the ANTLER research team. To gain access, researchers can sign a data access agreement with the study sponsor (priment@ucl.ac.uk, University College London, London, UK) and data can then be made available with support from the ANTLER research team.”

4. Is the manuscript presented in an intelligible fashion and written in standard English?

Reviewer #1: No

Response: We have made a number of changes which we hope will improve the reader’s understanding: including adding a missing reference on page 19 and a new reference on page 7; correcting typos on pages 15 and 16; correcting grammatical errors on page 19; replacing the headings “relapsed at 12 weeks” with “relapsed by 12 weeks” in the Tables 4-6 on pages 15 and 16

5. Review Comments to the Author

Reviewer #1: Line 130: rCIS-r were measured at 12, 26, 39 and 52 weeks.

Line 138: The rCIS-R was …. asked about the previous 12 weeks.

This is confusing. Is the measure all about previous 12 weeks, even though it is measured at 26, 39 and 52 weeks?

Response: that is correct at the follow-ups (12, 26, 39 and 52 weeks) the participants were asked to recall previous 12 weeks. We have added the text to emphasise the point that the follow ups in the trial were about 13 weeks apart on average and the rCIS-R measured recall in the previous 12 weeks. The Measure sub-section of Methods section on page 7 now reads: 

“The rCIS-R was a modified version of CIS-R and designed as a self-administered computerised questionnaire and asked about the previous 12 weeks(18). Of note, the follow ups in the trial were about 13 weeks apart on average.”

Did all 396 participants have outcome measures at every time point?

Response: We have added extra text to clarify this point, the Sample characteristic sub-section on page 10: “For the purpose of our study, 396 participants from the ANTLER trial (n=478) completed the rCIS-R twice at one of the four follow-ups.” 

If rCIS-r, GRQ and PHQ9 were measured at all time points, why did the analysis focus on 12 weeks only?

Response: The main outcome of the ANTLER trial was time to relapse with the 12-week follow-up being of primary interest due to potential overlap with withdrawal symptoms. Our aim was to assess the reliability and validity of the measure used to assess the main outcome. Therefore, this paper is using data from when the rCISR was repeated to enable the reliability to be estimated. We have added additional text to the manuscript to make this clearer (page 7).

The primary outcome is defined as time to relapse. Survival analysis by randomization group? No data analysis is performed for this outcome except the reliability by week or month which is not clearly described.

Response: We have made a number of changes to make a clear distinction between the main trial paper and the current manuscript (pages 6, 7, 10 and 14). The primary outcome of the ANTLER trial was time to relapse, survival analysis was performed, the results are published in the trial paper https://www.nejm.org/doi/full/10.1056/NEJMoa2106356. The purpose of the study in this paper was to assess reliability and validity of the measure (rCIS-R) that was used in the ANTLER trial to assess primary outcome. In our study, we treat participants who provided the data as a single cohort regardless of group allocation We have added additional text in the paper to make this clear (pages 6, 7, 10 and 14). 

Table 4 and 5, how did you define relapse if there is a disagreement between 1st and 2nd completion?

Table 4 and 5, how about other time points? P value to table 5?

Response: the purpose of 1st and 2nd completion was to assess the reliability of the measure used in the main trial to define relapse with the 12-week follow-up being of primary interest. We have added p-value to Table 5 (page 16). For this paper we are not concerned with how we defined the primary outcome for the main trial, the results of which are already published. 

Table 6, correlation coefficient or standardized regression coefficient are better to be reported as HPH9 and rCIS-r have different units. Add p values.

Response: thank you for the suggestions, we have added to Table 6 the standardised regression coefficients and p-values (page 6). 

“Excellent” is a too strong word for a kappa of 0.84.

Response: thank you for pointing that out, we have replaced “excellent” with “substantial” throughout the manuscript (pages 2, 12, 13 and 16). We followed classification suggested by Chmura Kraemer, H., Periyakoil, V.S. and Noda, A. (2002), Kappa coefficients in medical research. Statist. Med., 21: 2109-2129. https://doi.org/10.1002/sim.1180

6. PLOS authors have the option to publish the peer review history of their article (what does this mean?). If published, this will include your full peer review and any attached files.

Do you want your identity to be public for this peer review? For information about this choice, including consent withdrawal, please see our Privacy Policy.

Reviewer #1: No

---

## [Decision Letter · Decision Letter 1]

22 Nov 2022

PONE-D-22-01963R1Reliability of the retrospective Clinical Interview Schedule Revised (rCIS-R) to assess relapse in depression in primary care patients.PLOS ONE

Dear Dr. Duffy,

Thank you for submitting your manuscript to PLOS ONE. After careful consideration, we feel that it has merit but does not fully meet PLOS ONE’s publication criteria as it currently stands. Therefore, we invite you to submit a revised version of the manuscript that addresses the points raised during the review process.

 Your manuscript has been assessed by two reviewers whose reports can be found below. As you will see from the comments, reviewer #2  raised a number of concerns that need attention. They request additional information on methodological aspects of the study. Could you please carefully revise the manuscript to address all comments raised? Please submit your revised manuscript by Jan 06 2023 11:59PM. If you will need more time than this to complete your revisions, please reply to this message or contact the journal office at plosone@plos.org. Please include the following items when submitting your revised manuscript:A rebuttal letter that responds to each point raised by the academic editor and reviewer(s). You should upload this letter as a separate file labeled 'Response to Reviewers'.A marked-up copy of your manuscript that highlights changes made to the original version. You should upload this as a separate file labeled 'Revised Manuscript with Track Changes'.An unmarked version of your revised paper without tracked changes. You should upload this as a separate file labeled 'Manuscript'.If applicable, we recommend that you deposit your laboratory protocols in protocols.io to enhance the reproducibility of your results. Protocols.io assigns your protocol its own identifier (DOI) so that it can be cited independently in the future. For instructions see: https://journals.plos.org/plosone/s/submission-guidelines#loc-laboratory-protocols. Additionally, PLOS ONE offers an option for publishing peer-reviewed Lab Protocol articles, which describe protocols hosted on protocols.io. Read more information on sharing protocols at https://plos.org/protocols?utm_medium=editorial-emailutm_source=authorlettersutm_campaign=protocols.

We look forward to receiving your revised manuscript.

Kind regards,

Katrien Janin

Staff Editor

PLOS ONE

Journal Requirements:

Reviewers' comments:

Reviewer's Responses to Questions

**Comments to the Author**

1. If the authors have adequately addressed your comments raised in a previous round of review and you feel that this manuscript is now acceptable for publication, you may indicate that here to bypass the “Comments to the Author” section, enter your conflict of interest statement in the “Confidential to Editor” section, and submit your "Accept" recommendation.

Reviewer #1: All comments have been addressed

Reviewer #2: (No Response)

2. Is the manuscript technically sound, and do the data support the conclusions?

Reviewer #1: (No Response)

Reviewer #2: Partly

3. Has the statistical analysis been performed appropriately and rigorously? 

Reviewer #1: (No Response)

Reviewer #2: Yes

4. Have the authors made all data underlying the findings in their manuscript fully available?

Reviewer #1: (No Response)

Reviewer #2: Yes

5. Is the manuscript presented in an intelligible fashion and written in standard English?

Reviewer #1: (No Response)

Reviewer #2: Yes

6. Review Comments to the Author

Reviewer #1: (No Response)

Reviewer #2: Review comments on the manuscript titled "Reliability of the retrospective Clinical Interview Schedule Revised (rCIS-R) to assess relapse in In the study, the authors showed resutls of test-retest reliability of the rCIS-R to assess relapse in primary care patients.

In the study, the authors showed results of test-retest reliability of the rCIS-R as a secondary analysis of an RCT. And additionally, they examined the association between rCIS-R and other self-rated ratings (PHQ-9 and GRQ).

The manuscript has been reviewed by other reviewers and some comments have already been addressed by the authors. To the comments and responses, I have no additional comments.

However, I have other comments as below.

Major comments

My first concern is the lack of a clear demonstration of validity. Reliability is confirmed by the retest method. The validity of the CIS-R, which has already been validated, may have construct validity by using the same questions as the original CIS-R, but it cannot be validated by examining the relationship with the PHQ-9 and the GRQ.

Although the authors declare that they did an exploratory study, it may be an oversimplification to argue that there may be a little validity by adding that exploratory data.

The PHQ-9 measures status over the past 2 weeks, which differs from the rCIS-R in the time period of the measurement. It does not reflect results from the past 12 weeks.

The GRQ is a self-report of a broader disorder that is not indicative of depression, and the duration of the measure differs from the rCIS-R. It may not be sufficient for the external validity of the rCIS-R for the purpose of measuring recurrence of depression.

The authors may have positioned the validity study as only an exploratory study in the present study. In fact, the title of the paper says "reliability" but not "validity. However, on page 6, line 120, it says "Our aim was to assess the reliability and validity. This should be a cautionary statement because the validity that can be examined with the current method is highly constrained.

In addition, since this study is a secondary analysis incorporated into the original randomized controlled trial, its study design is difficult to understand. Other reviewers may have already noted this point.

In particular, the description of when the rCIS-R was conducted is difficult to understand. Table 1 shows that rCIS-R was performed at 12, 26, 39, or 52 weeks, and most of them were performed at 26 weeks. However, the description in the text does not tell us how this implementation collection was determined. Ideally, it would be better to assign them equally to each timing, or to set a precedent at some point to prevent bias.

Minor comments

It would be good if you could explain somewhere why you decided to evaluate the past 12 weeks and the reasonableness of your decision.

As mentioned above, on page 6, line 118, it is not clear whether "The primary outcome" refers to the outcome of the original RCT or this accompanying study to confirm reliability. It is difficult to tell whether "the primary outcome" refers to the original RCT or to this accompanying study to confirm reliability.

The description in Table 1 is confusing as to whether it is "mean" or "n".

On page 16, line 275, there is the phrase "strong association," but it is unclear on what basis it is described as "strong. The result may not be a very strong association.

7. PLOS authors have the option to publish the peer review history of their article (what does this mean?). If published, this will include your full peer review and any attached files.

Reviewer #1: No

Reviewer #2: No

---

## [Author Response · Author response to Decision Letter 1]

16 Dec 2022

Dear Dr Katrien Janin,

Re: PONE-D-22-01963 

Reliability of the retrospective Clinical Interview Schedule Revised (rCIS-R) to assess relapse in depression in primary care patients.

PLOS ONE

Thank you for arranging the second reviewer and for the invitation to submit a revised version of our manuscript. The main aim of our study was to assess the reliability of the rCIS-R. However, we also conducted exploratory analyses of construct validity to enhance our reliability finding and we are grateful to the reviewer for bringing our attention to it. We realised that the term “validity” could be misinterpreted as “external validity”, which is a different approach towards establishing validity. As a result, we clarified that we were addressing construct validity throughout the manuscript improving its quality. We provide a point-by-point response to the editor’s and reviewer’s comments below. The changes are highlighted in the response below.

We have also included: 

• A marked-up copy of the manuscript that highlights changes made to the original version. This is uploaded as a separate file labelled 'Revised Manuscript with Track Changes'.

• An unmarked version of our revised paper without tracked changes, uploaded as a separate file labelled 'Manuscript'.

Yours sincerely,

Larisa Duffy 

UCL Division of Psychiatry

Maple House, 149 Tottenham Court Road

London W1T 7NF

 

Editors’ comments

PONE-D-22-01963R1

Reliability of the retrospective Clinical Interview Schedule Revised (rCIS-R) to assess relapse in depression in primary care patients.

PLOS ONE

Journal Requirements:

Authors response: we have made corrections to the reference list

Reviewers' comments:

Reviewer's Responses to Questions 

Comments to the Author

1. If the authors have adequately addressed your comments raised in a previous round of review and you feel that this manuscript is now acceptable for publication, you may indicate that here to bypass the “Comments to the Author” section, enter your conflict of interest statement in the “Confidential to Editor” section, and submit your "Accept" recommendation.

Reviewer #1: All comments have been addressed

Reviewer #2: (No Response)

Authors’ response: thank you; we are glad to learn that reviewer#1 accepted all our responses to their comments.

2. Is the manuscript technically sound, and do the data support the conclusions?

Reviewer #1: (No Response)

Reviewer #2: Partly

Authors’ response: we have made a number of changes (listed below) to the manuscript.

3. Has the statistical analysis been performed appropriately and rigorously? 

Reviewer #1: (No Response)

Reviewer #2: Yes

Authors’ response: thank you

4. Have the authors made all data underlying the findings in their manuscript fully available?

Reviewer #1: (No Response)

Reviewer #2: Yes

Authors’ response: thank you

5. Is the manuscript presented in an intelligible fashion and written in standard English?

Reviewer #1: (No Response)

Reviewer #2: Yes

Authors’ response: thank you

6. Review Comments to the Author

Reviewer #1: (No Response)

Reviewer #2: Review comments on the manuscript titled "Reliability of the retrospective Clinical Interview Schedule Revised (rCIS-R) to assess relapse in In the study, the authors showed resutls of test-retest reliability of the rCIS-R to assess relapse in primary care patients.

In the study, the authors showed results of test-retest reliability of the rCIS-R as a secondary analysis of an RCT. And additionally, they examined the association between rCIS-R and other self-rated ratings (PHQ-9 and GRQ).

The manuscript has been reviewed by other reviewers and some comments have already been addressed by the authors. To the comments and responses, I have no additional comments.

However, I have other comments as below.

Major comments

My first concern is the lack of a clear demonstration of validity. Reliability is confirmed by the retest method. The validity of the CIS-R, which has already been validated, may have construct validity by using the same questions as the original CIS-R, but it cannot be validated by examining the relationship with the PHQ-9 and the GRQ.

Although the authors declare that they did an exploratory study, it may be an oversimplification to argue that there may be a little validity by adding that exploratory data.

The PHQ-9 measures status over the past 2 weeks, which differs from the rCIS-R in the time period of the measurement. It does not reflect results from the past 12 weeks.

The GRQ is a self-report of a broader disorder that is not indicative of depression, and the duration of the measure differs from the rCIS-R. It may not be sufficient for the external validity of the rCIS-R for the purpose of measuring recurrence of depression.

Authors’ response: we thank the reviewer for bringing attention to our exploratory analysis of construct validity. We have improved the manuscript and made it clear that our exploratory analysis was designed to assess construct validity. We made it clear now by adding extra test (“to explore construct validity”) on page 6 and (“construct validity”) on page 19. We agree with the reviewer that it would have been theoretically sound to assess the external validity of rCIS- R by administering the original CIS-R, that is already validated measure, on a weekly basis during the 12-week period. However, this method would have been resource intensive and off-putting for participants and we did not have access to such data. Instead of external validity, we examined construct validity. We reasoned that the fact rCIS-R using the same question as a validated measure CIS-R is already a good indicator that our new measure is a valid measure of depression, and we assessed construct validity by choosing three relevant indicators: PHQ-9, GRQ and stopping trial medication. We explain this in strength/limitations section, though we toned down our claims throughout by stressing that it was an exploratory analysis of construct validity.

The authors may have positioned the validity study as only an exploratory study in the present study. In fact, the title of the paper says "reliability" but not "validity. However, on page 6, line 120, it says "Our aim was to assess the reliability and validity. This should be a cautionary statement because the validity that can be examined with the current method is highly constrained.

Authors’ response: we have amended the text on page 6, so to stress that it was an exploratory analysis, and it now reads: “Our aim was to assess the reliability and to explore construct validity of the measure used to assess the main outcome”

In addition, since this study is a secondary analysis incorporated into the original randomized controlled trial, its study design is difficult to understand. Other reviewers may have already noted this point.

In particular, the description of when the rCIS-R was conducted is difficult to understand. Table 1 shows that rCIS-R was performed at 12, 26, 39, or 52 weeks, and most of them were performed at 26 weeks. However, the description in the text does not tell us how this implementation collection was determined. Ideally, it would be better to assign them equally to each timing, or to set a precedent at some point to prevent bias.

Authors’ response: we have added extra description on page 9 of how we determined the data collection of the repeated measure: “Our intention was for all 478 participants to complete the rCIS-R twice at 26-week follow-up appointment for consistency and simplicity of the fieldwork. However, if participants preferred to complete it at different follow-up appointment, we allowed it. As a result, …”

Minor comments

It would be good if you could explain somewhere why you decided to evaluate the past 12 weeks and the reasonableness of your decision.

Authors’ response: we have added the rationale for using the past 12 weeks and added the following text on page 6: “The 12 weeks recall period was chosen because the follow up appointments were roughly spaced at 3 months or 12 week intervals and it was convenient anchor point for participants to remember what has happened since they last met with the researcher. Also, the 12 week interval was considered as appropriate length for participants to remember.”

As mentioned above, on page 6, line 118, it is not clear whether "The primary outcome" refers to the outcome of the original RCT or this accompanying study to confirm reliability. It is difficult to tell whether "the primary outcome" refers to the original RCT or to this accompanying study to confirm reliability.

Authors’ response: to differentiate between the original RCT and our study, we have specified “The primary outcome of the main trial…” on page 6 

The description in Table 1 is confusing as to whether it is "mean" or "n".

Authors’ response: we put “mean” in brackets, so it is easy to differentiate from “n”, Table 1 page 10

On page 16, line 275, there is the phrase "strong association," but it is unclear on what basis it is described as "strong. The result may not be a very strong association.

Authors’ response: we have removed the word “strong” on page 16

---

## [Decision Letter · Decision Letter 2]

13 Jan 2023

Reliability of the retrospective Clinical Interview Schedule Revised (rCIS-R) to assess relapse in depression in primary care patients.

PONE-D-22-01963R2

Dear Dr. Duffy,

We’re pleased to inform you that your manuscript has been judged scientifically suitable for publication and will be formally accepted for publication once it meets all outstanding technical requirements.

Kind regards,

Joseph Donlan

Staff Editor

PLOS ONE

Additional Editor Comments (optional):

Reviewers' comments:

Reviewer's Responses to Questions

**Comments to the Author**

1. If the authors have adequately addressed your comments raised in a previous round of review and you feel that this manuscript is now acceptable for publication, you may indicate that here to bypass the “Comments to the Author” section, enter your conflict of interest statement in the “Confidential to Editor” section, and submit your "Accept" recommendation.

Reviewer #1: All comments have been addressed

Reviewer #2: All comments have been addressed

2. Is the manuscript technically sound, and do the data support the conclusions?

Reviewer #1: (No Response)

Reviewer #2: Yes

3. Has the statistical analysis been performed appropriately and rigorously? 

Reviewer #1: (No Response)

Reviewer #2: Yes

4. Have the authors made all data underlying the findings in their manuscript fully available?

Reviewer #1: (No Response)

Reviewer #2: Yes

5. Is the manuscript presented in an intelligible fashion and written in standard English?

Reviewer #1: (No Response)

Reviewer #2: Yes

6. Review Comments to the Author

Reviewer #1: (No Response)

Reviewer #2: The authors have appropriately addressed each of my previous peer review comments, and the text has been revised accordingly.

I have no new review comments to add.

7. PLOS authors have the option to publish the peer review history of their article (what does this mean?). If published, this will include your full peer review and any attached files.

Reviewer #1: No

Reviewer #2: No

---

## [Editor Report · Acceptance letter]

6 Mar 2023

PONE-D-22-01963R2 

Reliability of the retrospective Clinical Interview Schedule Revised (rCIS-R) to assess relapse in depression in primary care patients. 

Dear Dr. Duffy:

I'm pleased to inform you that your manuscript has been deemed suitable for publication in PLOS ONE. Congratulations! Your manuscript is now with our production department. 

Kind regards, 

on behalf of

Dr Joseph Donlan 

Staff Editor

PLOS ONE